# Comparative Mitogenome Analyses Uncover Mitogenome Features and Phylogenetic Implications of the Reef Fish Family Holocentridae (Holocentriformes)

**DOI:** 10.3390/biology12101273

**Published:** 2023-09-22

**Authors:** Qin Tang, Yong Liu, Chun-Hou Li, Jin-Fa Zhao, Teng Wang

**Affiliations:** 1College of Fisheries, Huazhong Agricultural University, Wuhan 430070, China; tangqin@mail.hzau.edu.cn; 2Key Laboratory of South China Sea Fishery Resources Exploitation and Utilization, Ministry of Agriculture and Rural Affairs, South China Sea Fisheries Research Institute, Chinese Academy of Fishery Sciences, Guangzhou 510300, China; liuyong@scsfri.ac.cn (Y.L.); chunhou@scsfri.ac.cn (C.-H.L.); zhaojf2019@163.com (J.-F.Z.); 3Scientific Observation and Research Station of Xisha Island Reef Fishery Ecosystem of Hainan Province, Key Laboratory of Efficient Utilization and Processing of Marine Fishery Resources of Hainan Province, Sanya Tropical Fisheries Research Institute, Sanya 572018, China; 4Guangdong Provincial Key Laboratory of Fishery Ecology Environment, Guangzhou 510300, China; 5Observation and Research Station of Pearl River Estuary Ecosystem, Guangzhou 510300, China

**Keywords:** Holocentridae, mitogenome, codon usage, gene rearrangement, phylogeny, selection pressure

## Abstract

**Simple Summary:**

Mitochondria play a critical role in the energy metabolism of coral reef fish, providing ATP to fuel cellular processes. A mitogenome study has been employed to investigate the genetic diversity, population structure, and evolutionary relationships among coral reef fish taxa. Species of the Holocentridae family play important ecological roles in coral reef communities. Two subfamilies of this family, Holocentrinae and Myripristinae, exhibit similarities in morphology and distribution, with minor differences in habitation and feeding behavior. Here, we present full mitochondrial genome sequences of eight holocentrid species and report the results of a comparative analysis with six previously published species. The results indicate that these mitogenome structures are relatively conserved, except for the high variability in control regions. The whole genomes, except for *nad6*, exhibited positive AT-skews and negative GC-skews. Furthermore, we compared the two subfamilies to explore the reasons behind their varying inhabitation and behavior. Phylogenetic analysis indicated all species formed two subfamilies, the Holocentrinae and Myripristinae, with each subfamily comprising two genera. Positive selection analysis revealed that all protein-coding genes (PCGs) were subjected to purifying selection. The data obtained from our study could serve as a valuable resource for future investigations on the evolution and conservation of holocentrid fish.

**Abstract:**

To understand the molecular mechanisms and adaptive strategies of holocentrid fish, we sequenced the mitogenome of eight species within the family Holocentridae and compared them with six other holocentrid species. The mitogenomes were found to be 16,507–16,639 bp in length and to encode 37 typical mitochondrial genes, including 13 PCGs, two ribosomal RNAs, and 22 transfer RNA genes. Structurally, the gene arrangement, base composition, codon usage, tRNA size, and putative secondary structures were comparable between species. Of the 13 PCGs, *nad6* was the most specific gene that exhibited negative AT-skews and positive GC-skews. Most of the genes begin with the standard codon ATG, except *cox1*, which begins with the codon GTG. By examining their phylogeny, *Sargocentron* and *Neoniphon* were verified to be closely related and to belong to the same subfamily Holocentrinae, while *Myripristis* and *Ostichthys* belong to the other subfamily Myripristinae. The subfamilies were clearly distinguished by high-confidence-supported clades, which provide evidence to explain the differences in morphology and feeding habits between the two subfamilies. Selection pressure analysis indicated that all PCGs were subject to purifying selection. Overall, our study provides valuable insight into the habiting behavior, evolution, and ecological roles of these important marine fish.

## 1. Introduction

Holocentridae, a family of ray-finned fish, is also known as a nocturnal coral reef fish family, with the subfamily Holocentrinae typically known as squirrelfish, while Myripristinae members are known as soldierfish [1,2,3]. The family Holocentridae is primarily distributed in the tropical parts of the Atlantic, Indian, and Pacific Oceans. Typically, they inhabit waters up to a depth of 100 m, although some species of the *Ostichthys* genus (soldierfish) from the subfamily Myripristinae have been detected much deeper [4]. Holocentridae fish possess large eyes and are primarily active at night, suggesting they are nocturnal in terms of their activity patterns [5]. The colors of the majority of Holocentridae are either red or silver [6]. Members of the Holocentrinae subfamily (squirrelfish) possess venomous spines near the gill opening, which can inflict painful wounds [7]. Regarding feeding habits, squirrelfish mainly feed on small fish and benthic invertebrates, while soldierfish typically feed on zooplankton [8]. Unlike adults, the larvae of Holocentridae are pelagic and can be found far out at sea [9]. Currently, according to the statistics of Fishes of the Word, a total of 83 species belonging to 8 genera are recognized, with *Sargocentron* and *Myripristis* being the most numerous genera, which contain 33 and 28 species, respectively [10]. Based on our investigation, only four genera of Holocentridae have been studied at the mitochondrial genome level so far, including *Sargocentron*, *Neoniphon*, *Myripristis*, and *Ostichthys*, with *Sargocentron* and *Neoniphon* belonging to the Holocentrinae subfamily and *Myripristis* and *Ostichthys* falling under the Myripristinae subfamily.

The Holocentridae family contributes to the ecological diversity of coral reefs. Prior to the disclosure of its complete mitochondrial genome structure, several studies concentrated on the physiology, ecology, and evolutionary aspects of this fish family. In a report by Eric et al. in 2011, Holocentrids were identified as vocal reef fishes. In their study, the authors compared sound production mechanisms across different species and found that all fish possess fast-contracting muscles and have relatively similar sound-producing mechanisms [1]. Fanny et al. (2021) studied the visual systems of Holocentridae and compared the two subfamilies, the Holocentrinae and Myripristinae, demonstrating that squirrelfish had a slightly more developed photopic visual system than soldierfish [3]. The evolution of holocentroids has also been evaluated. Andrews et al. (2023) reported a new holocentroid species from the fossil material of the early Paleocene and estimated a Danian divergence between Myripristinae and Holocentrinae from the fossil analysis via micro-computed tomography, suggesting that several holocentroid lineages crossed the Cretaceous–Palaeogene boundary [4]. These studies give us insight into the traits of holocentrid fish and their ability to adapt to the marine environment, as well as their evolutionary history and important role in the ocean ecosystem. Next, a study conducted at the mitochondrial structure level will provide proof of previous findings and reveal a systematic relationship of holocentrid species.

Mitochondria play a crucial role within eukaryotic cells, participating in various essential processes such as ATP generation through oxidative phosphorylation, cell differentiation, signaling, growth, and apoptosis [11,12,13]. The vertebrate mitogenome is characterized by its small size, ranging from 16 to 17 kb, and its circular double-stranded structure. A typical mitogenome contains 13 protein-coding genes (PCGs), 22 transfer RNA genes (tRNAs), 2 ribosomal RNA genes (rRNAs), and two non-coding regions, namely, the origin of L-strand replication (O_L_) and the control region (CR) [14,15]. Mitochondrial DNA sequences have been extensively studied across various fields. In evolutionary biology, they have been used to investigate the evolutionary relationships and genetic variation between different species, while in biogeography, they have been used to uncover spatial distributions and migration patterns [15]. Although mitochondrial function in corals is highly conserved, it has been observed that the PCGs of some species undergo evolutionary selection in response to the metabolic demands imposed by extreme environments [16,17,18,19]. In a recent study by Ramos et al. (2023), selection tests were conducted on mitochondrial PCGs of deep-sea and shallow-water species, revealing that certain PCGs underwent adaptive evolution during their adaptation to the deep-sea environment [16]. Another extreme environment, the sub-zero habitat of the Antarctic, poses a significant challenge to the survival of fish. Thus, Antarctic icefishes have developed a unique mechanism to adapt to the inhabitants. In an earlier study conducted in 2010, O’Brien et al. observed an increase in mitochondrial density in cardiac myocytes and oxidative skeletal muscle fibers, accompanied by a proliferation of mitochondrial membranes. This expansion of membranes facilitates the efficient intracellular diffusion of oxygen [20].

Although holocentrids play an irreplaceable role in coral reef ecosystems and several studies have focused on their morphological characteristics, activity patterns, and underlying mechanisms, there has been limited research on their mitogenome characteristics and evolutionary biology. In this study, we present eight mitogenomes for the first time and report the results of a comparative analysis with six published mitogenomes. We provide comprehensive insights into the detailed features of all mitogenomes with respect to structure, gene arrangement, nucleotide composition, noncoding RNA, and codon usage. Additionally, we explore the phylogenetic relationships between species and estimate the selection pressures during their evolution. Through comparative analysis and newly generated results, we offer valuable insights into the evolutionary history of holocentrid species. Furthermore, we make meaningful contributions towards identifying and protecting these coral reef fish species.

## 2. Materials and Methods

### 2.1. Sampling, DNA Extraction, Library Construction, and Sequencing

In this study, we de novo sequenced eight holocentrid species, including *Myripristis kuntee* (Shoulderbar soldierfish), *Myripristis murdjan* (Pinecone soldierfish), *Myripristis violacea* (Lattice soldierfish), *Neoniphon opercularis* (Blackfin squirrelfish), *Sargocentron caudimaculatum* (Silverspot squirrelfish), *Sargocentron diadema* (Crown squirrelfish), *Sargocentron melanospilos* (Blackblotch squirrelfish), and *Sargocentron punctatissimum* (Speckled squirrelfish). The specimens were obtained from the Xisha Islands (15°46′~17°08′ N, 111°11′~112°54′ E), China, and deposited in the South China Sea Fisheries Research Institute, Chinese Academy of Fishery Sciences. Six published mitogenome sequences from the other six holocentrid species were downloaded from GenBank for an integrative and comparative analysis: *Neoniphon samara* (Sammara squirrelfish), NC_063501.1; *Sargocentron spiniferum* (Sabre squirrelfish), KX254549.1; *Sargocentron rubrum* (Redcoat; squirrelfish), NC_004395.1; *Myripristis vittate* (Whitetip soldierfish), NC_063496.1; *Myripristis berndti* (Blotcheye soldierfish), AP002940.1; *Ostichthys japonicus* (Japanese soldierfish), AP004431.1.

Total genomic DNA was extracted from the specimens using the E.Z.N.A.^®^ Tissue DNA Kit (OMEGA, Beijing, China) in accordance with the manufacturer’s protocols. Two distinct types of tissue were sampled: a small fragment of muscle from a specimen or a clip of the pelvic fin taken from the right side of a specimen. After DNA extraction, 1 μg of purified DNA was randomly fragmented into fragments with a length ranging from 300 to 500 bp and used for subsequent library construction. Complete genomic libraries were established using the Illumina TruSeqTM Nano DNA Sample Prep Kit (Illumina, San Diego, CA, USA) in accordance with the manufacturer’s instructions. Then, libraries were sequenced using the Illumina NovaSeq 6000 platform to obtain 150 bp paired-end reads. The library construction and sequencing procedures were carried out by the Biozeron Corporation (Shanghai, China).

### 2.2. Sequence Assembly, Annotation, and Analyses

Prior to assembly, raw reads were filtered using Trimmomatic (v0.39) [21] to remove the low-quality reads (the reads showing a quality score below 20, Q < 20), the reads with adaptors, the reads containing a percentage of uncalled bases (“N” characters) equal to or greater than 10%, and duplicated sequences. Filtered reads were assembled into contigs using MitoZ (v2.3) [22], and potential mitochondrial contigs were extracted by aligning them against the NCBI mitogenome database. Then, GetOrganelle (v1.7.5) (https://github.com/Kinggerm/GetOrganelle, accessed on 20 March 2022) was used to assemble the mitogenomes [23]. After assembly, the starting position and orientation of the mitochondrial sequence were obtained based on a reference genome. Annotation of the mitogenomes to protein-coding genes (PCGs), tRNAs, and rRNAs was performed using MITOS [24] and Mitoannotator (v3.83) [25]. Functional annotations of PCGs were performed using sequence-similarity Blast searches with a typical cut-off E-value of 10^−5^ against several publicly available protein databases: NCBI non-redundant (Nr) protein database, Swiss-Prot, Clusters of Orthologous Groups (COGs), and Kyoto Encyclopedia of Genes and Genomes (KEGG) and Gene Ontology (GO) terms. tRNA genes were searched using tRNAscan-SE (v2.0) [26], and their secondary structures were drawn via RNAplot from the package ViennaRNA (v2.5.1) [27]. Base composition and codon distributions were analyzed using MEGA 7.0 [28], and the nucleotide composition skewness was measured using the following formulas: AT-skew = (A − T)/(A + T) and GC-skew = (G − C)/(G + C) [15]. Relative synonymous codon usage (RSCU) was calculated using the “cusp” of EMBOSS package (v6.6.0.0) [29]. Circular genomes were visualized using the CGView tool (http://stothard.afns.ualberta.ca/cgview_server/, accessed on 25 March 2022) online [30]. The r package “ComplexHeatmap” was used to draw heatmaps. The conserved-sequence block domains (CSBs) of control regions were determined by comparing them with public holocentrid species.

### 2.3. Phylogenetic Analyses

The phylogenetic relationships were reconstructed using the 13 PCGs of the 14 holocentrid fish mitogenomes; three parrotfish, namely, *Scarus frenatus* (OQ349185.1, Bridled parrotfish), *Scarus niger* (OQ349187.1, Dusky parrotfish), and *Scarus prasiognathos* (OQ349189.1, Singapore parrotfish), were used as outgroup taxa. Multiple sequence alignment was performed using MAFFT (v7.453) [31] with default parameters. The alignment results were further trimmed to eliminate the ambiguous positions using Gblocks (v0.91b) [32]. Then, trimmed sequences were concatenated into a supermatrix with FASconCAT [33]. Fasta files were converted into Nexus format using Geneious (v.2022.2.2) [34]. Phylogenetic relationships were inferred from the concatenated dataset using maximum likelihood (ML) and Bayesian inference (BI) methods. ML analyses were performed using IQ-TREE (v1.6.12) [35] with the following parameters: “-m MFP -b 1000 -bnni”. By using these parameters, the best-fit substitution models (including FreeRate heterogeneity models) and partition schemes were inferred via the built-in ModelFinder [36]. “MFP” (ModelFinder Plus) allows for extended model selection followed by tree inference, while “-b 1000” ensures that 1000 bootstrap searches will be performed in order to infer the consensus trees. BI analysis was carried out using MrBayes (v3.2.7a) [37]. Before BI analysis, the best model was selected with jModeltest (v2.1.10) [38], and the model of JC was optimal for analysis with nucleotide alignment. Then, BI analysis was performed using four simultaneous Markov Chain Monte Carlo chains for 2,000,000 generations and sampled every 1000 generations, using a burn-in of 25% generations. The average standard deviation of split frequencies was set as less than 0.01. Phylogenetic trees generated from both ML and BI methods were visualized in FigTree (v1.4.4) (http://tree.bio.ed.ac.uk/software/figtree/, accessed on 10 June 2023).

### 2.4. Positive Selection

Positive selection refers to the evolutionary process through which genetic variants increase in frequency within a given population until they become prevalent. This phenomenon results from the advantageous traits conferred by these genetic variants, which enhance the fitness and reproductive success of the individuals carrying them. To perform positive selection analysis, first, a multiple-codon alignment was produced for each PCG from the corresponding aligned predicted protein sequences using PAL2NAL [39]. Then, positive selection analyses were performed using two codon-based maximum likelihood methods, i.e., Single Likelihood Ancestor Counting (SLAC) and Fixed Effects Likelihood (FEL), as implemented via HYPHY (v2.5.39) (MP) [40] on a Linux system. SLAC represents a substantially enhanced and refined version of the Suzuki–Gojobori counting method and is designed to assess the rate of nonsynonymous and synonymous substitutions in DNA sequences, thereby shedding light on the selective pressures operating on specific genes during the progress of evolution. FEL is also an innovative and statistically robust approach rooted in likelihood-based methods aiming to characterize the evolutionary dynamics of genetic sequences in the context of codon substitution models. In this study, the number of non-synonymous substitutions per non-synonymous site (dN) and the number of synonymous substitutions per synonymous site (dS) were estimated using both methods. And the dN/dS ratio (or ω) was taken as a judgment of the selective pressure on each codon of the PCGs. In detail, the ratio dN/dS > 1 suggests positive or diversifying selection, dN/dS < 1 suggests negative or purifying selection, and dN/dS = 1 indicates neutral evolution. The significance level of the positive selection estimated from both SLAC and FEL analyses was set as *p*-value < 0.05.

## 3. Results

### 3.1. General Features of Mitochondrial Genomes

The total length of the eight newly sequenced complete mitogenomes ranged from 16,507 bp in *Sargocentron punctatissimum* to 16,639 bp in *Neoniphon opercularis* (Appendix A). All mitogenomes comprised 37 genes, including 13 PCGs, two rRNAs (12S rRNA and 16S rRNA, named *rrnS* and *rrnL*), one control region (CR), and 22 tRNAs (Table 1, Figure 1, Appendix A, Appendix A). Taking *Myripristis kuntee* as an example, the total length of the 13 PCGs in the mitogenome is 11,439 bp, which accounts for 69.20% of the entire mitogenome. In total, 12/13 of the PCGs are encoded on the Heavy (H) strand in the positive direction, except for the gene *nad6* (NADH dehydrogenase subunit 6), which is located on the Light (L) strand in the reverse direction (Figure 1, Table 1). Fourteen of the tRNAs, namely, *trnD, trnK, trnG, trnR, trnH, trnS1, trnL1, trnT, trnF, trnV, trnL2, trnI, trnM,* and *trnW,* are located on the H-strand, while the other eight tRNAs (*trnS2, trnE, trnP, trnQ, trnA, trnN, trnC,* and *trnY*) are located on the L-strand (Figure 1, Table 1). This arrangement pattern of genes is identical among holocentrid species (Appendix A) and is similar to that found in most vertebrates [41]. The comparative analysis of 14 mitochondrial genomes with respect to structure reveals that they are almost identical, and no rearrangements of genes have occurred, but the control region was the most variable region among the species in both length and nucleotide composition (Figure 2, Appendix A).

Two types of start codons and five types of stop codons were used in 14 species. Taking *M. kuntee* as an example, most of the PCGs begin with the standard codon ATG, except *cox1* (cytochrome c oxidase subunit I), which begins with the codon GTG (Table 1). Table 1 also displays the utilization of stop codons in the *M. kuntee* mitochondrial genome. Overall, seven PCGs (*atp8*, *atp6, cox3*, *nad4l*, *nad5*, *nad1*, and *nad2*) end with TAA, while *nad3* terminates with TAG, *nad6* terminates with AGG, *cox1* terminates with AGA, and the remaining three PCGs (*cox2*, *nad4*, and *cob*) end with an incomplete terminating codon T-- (Table 1).

### 3.2. Nucleotide Composition of Protein-Coding Genes of the Holocentrid Mitogenomes and the Codon Usage

The nucleotide compositions were comparable among all 14 species. For the whole mitochondrial genome, the overall A + T content ranges from 53.12% in *M. murdjan* to 56.85% in *N. sammara*, while the G + C content ranges from 43.15% in *N. sammara* to 46.88% in *M. murdjan* (Appendix A). For the PCGs of the 14 mitogenomes, the average A + T content is 53.95%, which is slightly lower than the average of the whole genome, 54.70%. When focusing on each of the 13 PCGs, the lowest A + T content was found in *nad4l* (50.63 ± 1.50%), while the highest was found in *cox2* (55.33 ± 1.61%) (Appendix A). Taking *M. kuntee* as an example, the A + T content of its PCGs was 52.20% (Appendix A). All the holocentrid mitogenomes exhibited AT bias in the whole mitogenome, tRNAs, rRNAs, and most of the PCGs (Figure 3, Appendix A). The largest AT-skew values were observed in rRNAs, all of which were positive, while the smallest and most negative values were found in the PCG nad6 (Figure 3, Appendix A). For most of the PCGs, the AT-skew was higher than the GC-skew, except for nad6, which exhibited an unusual AT-skew and GC-skew (Figure 3). The negative AT-skew and positive GC-skew observed indicated that nad6 displayed an excess of T over A and G over C. Moreover, the average AT-skew and GC-skew values of all 13 PCGs were negative (Figure 3, Appendix A).

The relative synonymous codon usage (RSCU) values for the PCGs are summarized in Figure 4 and Appendix A. Excluding stop codons, there are 3813 codons in the mitogenome of *M. kuntee*. The codons encoding Arg, Leu, and Ser are the most frequent, while those encoding Trp and Met are scarce (Figure 4A). The heatmap, which was generated based on RSCU values, illustrates the resemblance of codon usage patterns among 14 mitogenomes (Figure 4B). Among the codons coding Ala, GCC (RSCU = 1.75) is the most frequently used. Also, it is the most frequently used codon among the 61 codons that encode amino acids.

### 3.3. Transfer RNA and Ribosomal RNA

All 22 typical tRNAs of the vertebrate mitochondrial genome were found in the mitogenomes (Figure 5A). Taking *M. kuntee* as an example, the tRNA size ranged from 65 to 74 bp. Most tRNAs could be folded into the canonical clover leaf secondary structure. The secondary structure of tRNAs generally contains four domains and a short variable loop: the amino acid acceptor (AA) stem, the dihydrouridine arm (D stem and loop, D), the thymidine arm (T stem and loop, T), the anticodon arm (AC stem and loop, AC), and the variable (V) loop (Figure 5A). However, as determined from the comparison of four representative species from the genera *Myripristis, Neoniphon, Sargocentron,* and *Ostichthys*, *trnC*-GCA (Cys) is supposed to lose the D loop in *M. kuntee* and *O. japonicus* (Figure 5B). The tRNA *trnS1*-GCT (Ser) is characterized by a special V loop and a large ring structure in the D loop in three of the four typical holocentrid species, but *O. japonicus* does not have these two structures (Figure 5B). Moreover, all species lack a D stem in *trnS1*-GCT (Ser), thus leading to failure in the formation of the typical clover leaf structure (Figure 5B). Regarding another special tRNA, *trnH*-GTG (His), when focusing on the T stem, *N. opercularis* and *S. caudimaculatum* possess a small ring due to the high GC content (Figure 5B).

Two ribosomal RNAs (12S rRNA and 16S rRNA, or *rrnS* and *rrnL*) are located on the H-strand in all holocentrid mitogenomes. These two genes were separated by *trnV*-TAC (Val), a feature often found in the mitochondrial genomes of vertebrates [41,42,43]. Taking *M. kuntee* as an example, the lengths of its 12S rRNA and 16S rRNA genes were 949 bp and 1645 bp, respectively. The total A + T content of these two rRNA genes was 53.93%, higher than their G + C content. Moreover, this mitogenome had a positive AT-skew (0.27) and a negative GC-skew (−0.12) (Appendix A). Other species showed similar results to *M. kuntee*.

### 3.4. Overlaps and Control Regions

When focusing on the 12 protein-coding genes on the heavy strand in the positive direction, a total of three overlaps from genes were detected in the mitogenomes of holocentrids (Table 1, Appendix A). Taking *Myripristis kuntee* as an example, the longest overlap was found between *atp8* and *atp6*, with a highly conserved 10 bp motif of “AGCTTCTTCG”, while the second longest overlap was found between *nad4l* and *nad4*, with a 7 bp sequence, “ATGCTAA”. Apart from that, a 5 bp overlapped sequence, “CCTAA”, was observed between *nad5* and *nad6*. Also on the same strand, the control region, located between *trnP* and *trnF*, with a range from 838 bp in *Sargocentron diadema* to 960 bp in *Neoniphon opercularis*, was the most variable region among species. Notably, this variable region accounted for the predominant portion of length discrepancies observed within the mitogenomes of holocentrids. Five conserved sequence blocks (CSB), CSB-I, CSB-II, CSB-III CSB-IV, and CSB-V, were detected (Figure 6) from the alignment of control regions. The base composition was extremely unique to each CSB, with CSB-I being T- and A-rich, CSB-II being AT- and C-rich, CSB-III being T- and C-rich, CSB-IV being C-rich, and CSB-V being A- and C-rich (Table 2).

On the light strand, the special non-coding region O_L_ (the origin of light strand replication), with a length ranging from 28 to 37 bp among species, is known to regulate the encoding of the *nad6* gene and eight tRNAs. Structurally, O_L_ is situated within the cluster of five tRNA genes (WANCY), and its secondary structure exhibits a stable stem-loop configuration, characterized by a tight structure with seven G-C pairs in the genera *Neoniphon*, *Sargocentron*, and *Ostichthys* and eight G-C pairs in the genus *Myripristis* (Figure 7). The G-C base pairs forming the stem exhibited a high level of conservation, maintaining their stability across different instances. In contrast, the composition of bases within the loop region displayed variability, with the presence of T being notably limited.

### 3.5. Phylogenetic Analysis

To explore the evolutionary patterns of the 14 holocentrid species, phylogenetic trees were constructed using both ML and BI methods based on 13 PCGs, and three parrotfish from our previous study were used as outgroups [15]. The topological structures of the phylogenetic trees obtained using the two methods were congruent, except that the BI tree had higher support values on the clade of *Neoniphon* (Figure 8). Both trees delimited two prominent clades: clade A and clade B. Clade A consists of species from the genera *Ostichthys* and *Myripristis*, while clade B consists of the other two genera, *Neoniphon* and *Sargocentron*. In general, different species of the same genus clustered into the same clade. In clade B, the genera *Neoniphon* and *Sargocentron* clustered together, with BI posterior probabilities (PP) equal to 1 and an ML bootstrap (BP) equal to 100, which implies a close relationship phylogenetically. However, the fact that the *Neoniphon* clade has become independent suggests it is a new population or a close relative of the genus *Sargocentron*. *O.japonicus* in clade A clustered together with species of *Myripristis*, suggesting its morphological similarity with that genus. Interestingly, two subfamilies of Holocentridae, Holocentrinae and Myripristinae, were clearly distinguished with high confidence, with a PP equal to 1 and a BP equal to 100, suggesting the independence of the squirrelfish and the soldierfish groups and potential differences in morphology and feeding habits between these two subfamilies.

### 3.6. Non-Synonymous, Synonymous Substitutions, and Positive Selection

To better understand the role of selective pressure, the dN and dS values of the PCGs were calculated using two codon-based maximum likelihood methods, SLAC and FEL. A total of 3381 amino acid sites was calculated using both methods (Figure 9). As determined from the SLAC result, 378 sites are prone to undergoing positive/diversifying selection (dN − dS > 0), but the *p*-values were not significant (>0.05). Of the rest 3003 sites, 1784 are under negative/purifying selection, with dN − dS < 0 and a *p*-value < 0.05 (Figure 9A), while the others have a *p*-value > 0.05. From the FEL analysis, 2459 sites were detected with dN/dS < 1 (ω < 1) (Appendix A). *nad5* and *atp8* presented the highest and lowest numbers of amino acids under purifying selection, respectively (Figure 9B), while *nad4* presented the highest percentage of amino acids under purifying selection (Figure 9C). Collectively, all PCGs were subject to purifying selection, with most dN − dS values lower than 0 or dN/dS values lower than 1 (ω < 1), taking a *p*-value < 0.05 as a threshold.

## 4. Discussion

In this study, we found that the overall codon usage among the 14 holocentrid species is similar. But when focusing on the PCGs within each species, most of the genes begin with the standard codon ATG, except *cox1*, which begins with the codon GTG. Previous research reported that ATG was the most prevalent in the mitochondrial genome of vertebrates and was exclusively used in the *cox3* gene, while GTG was primarily utilized in the *cox1* gene in over 95% of species [14]. In fish, such as the fathead minnow (*Pimephales promelas*) and parrotfishes, ATG acts as the start codon for all PCGs except *cox1*, which uses GTG as the start codon [15,44]. Unlike bony fish, in certain marine animals, like sea cucumbers, the start codon GTG is frequently employed in the genes responsible for encoding NADH dehydrogenase subunits, including genes like *nad1*, *nad4l*, and *nad5* [45]. Stop codon usage of the mitogenomes was also found to be similar among species, with more than half of the PCGs (7/13) using TAA as a stop codon; three PCGs terminating with TAG, AGG, and AGA, respectively; and the other PCGs ending with an incomplete terminating codon, i.e., T. The usage of start and stop codons in this study is comparable to findings for other fish. Satoh et al. (2016) conducted a codon usage analysis on 250 fish and discovered the utilization of nine types of start codons and seven types of stop codons [14]. The most frequently used start codons were ATG and GTG, whereas TAA, TAG, AGA, and AGG were all used as complete stop codons. Additionally, three types of incomplete stop codons (TA-, T--, and AG-) were also employed.

The RSCU analysis revealed that the codons were more prone to using A and T than C and G. The codons encoding Arg, Leu, and Ser were highly abundant, whereas those encoding Trp appeared infrequently (Figure 4A). These results are similar to those on the codon usage in the Characidae family [46] but different from this usage for invertebrate species such as *Lysmata vittate* [47]. Moreover, GCC-Ala (RSCU = 1.75) is the most frequently used codon in the mitogenome of *M. kuntee*. Usually, the RSCU intuitively reflects the preference for codon usage [48]. The observed bias of A and T nucleotides in holocentrid species likely contributed to a corresponding bias in the usage of codons. Previous studies reported that a notable characteristic of the mitochondrial genome in other teleost species is the A and T bias, resulting in a consequential bias in the encoded amino acids [14,15].

The comprehensive phylogenetic tree generated using the mitogenomes of holocentrid fish indicates the evolutionary position of the two subfamilies. The subfamilies distinctly diverged, which was supported with high confidence, providing evidence explaining the differences in morphology and feeding habits between the two subfamilies. According to previous studies, fish from these two subfamilies share some similarities but also demonstrate unique characteristics in their habitats and lifestyles. During the larval stage, both subfamilies live in the upper pelagic ocean and feed on zooplankton [49]. As they transition into juvenile life, the majority of holocentrids migrate to shallow tropical coral reef habitats [10], where the subfamily Holocentrinae adopts a nocturnal lifestyle and feeds on benthic crustaceans, while the subfamily Myripristinae feeds on zooplankton in the water column [50]. Our previous study on parrotfish indicated that ecological differences in habitats affect the formation of morphology and feeding habits and might act as the primary driving force in species diversification [15]. And the visual system of the two subfamilies differed after settlement, with Myripristinae showing a more pronounced adaptation for scotopic vision than Holocentrinae [2]. Moreover, high-confidence clades support the notion of two genera in each clade. Within Holocentrinae, a previous analysis indicated that there was strong support for the genera *Neoniphon* and *Sargocentron* being paraphyletic [51]. In more specific terms, the genome characteristics of the mitochondrion indicate that the species *N. opercularis* and *N. samara* are phylogenetically closer to *S. punctatissimum* and *S. diadema* than to other species within the genus *Sargocentron*. This could suggest the existence of an evolutionary correlation between the two genera. However, the cited author proposed that *Neoniphon* and *Sargocentron* probably underwent a complex evolution but did not derive from a common ancestry as determined via Bayesian ancestral state reconstruction methods [51].

Positive selection is an evolutionary process in which advantageous genetic variations (mutations) increase in frequency within a population. However, under natural conditions, the primary form of selection is purifying selection, which continually removes harmful mutations that occur in each generation [52,53]. In this study, the purifying selection for PCGs was prominent, thus ensuring that deleterious mutations cannot take over the population of Holocentridae. Previous studies on other reef fish have also shown that PCGs of the mitochondrial genome undergo purification selection [15], indicating fewer amino acid variations during evolution. The *nad5* is a core subunit of NADH dehydrogenase, which is located on the mitochondrial membrane and is involved in the function of the respiratory chain, while *atp8* synthesizes ATP and serves as the primary energy source for mitochondrial oxidative phosphorylation. The properties of purification exhibited by *nad5* and *atp8* make them potential markers for identifying holocentrids.

## 5. Conclusions

In this study, a mitogenome study was employed to investigate the genetic diversity, population structure, and evolutionary relationships of the reef fish family Holocentridae. Eight holocentrid mitogenomes were sequenced, and a comparative mitogenome analysis with six published holocentrid fish was performed. Our characteristic analysis indicated that a typical holocentrid mitogenome is 16,239 bp in length and encodes 37 genes. For all the species, the mitogenome structures were relatively conserved. The whole genomes of the mitochondria exhibited positive AT-skews and negative GC-skews. Among the 13 PCGs, *nad6* was the most specific gene that exhibited negative AT-skews and positive GC-skews. GCC-Ala is the most frequently used codon in the mitogenome. Most of the genes, except *cox1*, begin with the standard codon ATG. Our phylogenetic analysis supports the notion that the genera *Sargocentron* and *Neoniphon* belong to the subfamily Holocentrinae, while the genera *Myripristis* and *Ostichthys* belong to the subfamily Myripristinae. The clades provide evidence to explain the differences in morphology and feeding habits between different subfamilies. The conducted positive selection analysis indicates that all the PCGs were under purifying selection. *nad5* and *atp8* are potential markers for identifying holocentrids. Our study contributes to an in-depth understanding of the biological characteristics and evolutionary relationships of holocentrid fish and enriches the mitochondrial genome resources of coral reef fish.

## Figures and Tables

**Figure 1 biology-12-01273-f001:**
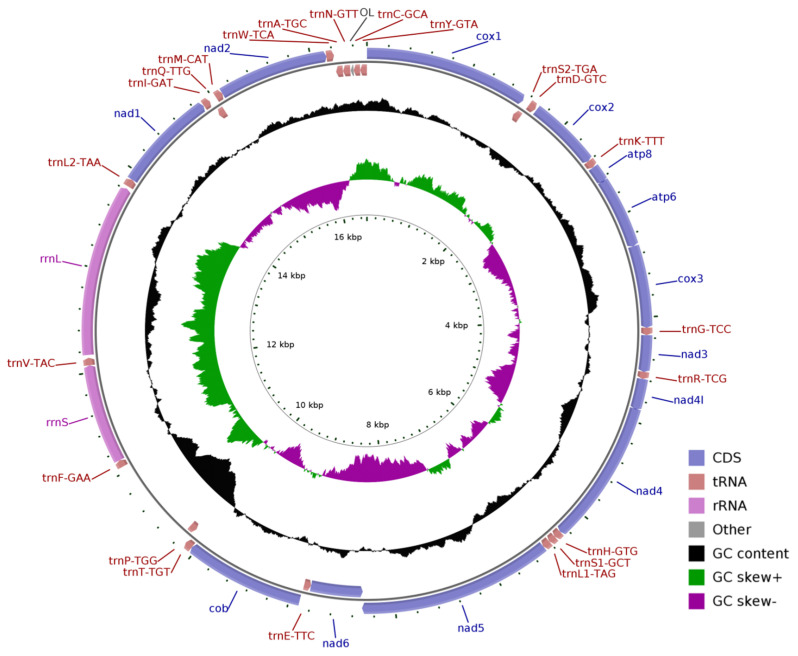
Map of the mitochondrial genomes of holocentrid species. *Myripristis kuntee* was taken as an example. PCGs are indicated by blue arrows, tRNA genes are indicated by brown arrows, and rRNA genes are indicated by lavender arrows. tRNAs are denoted by single-letter amino acid abbreviations followed by anticodons. Peaks on the black cycle indicate the GC content, while the outward and inward directions indicate GC content above or below average level. The purple and green cycles show the GC skew, where skew values between 0 and 1 are shown in purple and those between −1 and 0 are shown in green. Ticks in the inner cycle indicate the sequence length.

**Figure 2 biology-12-01273-f002:**
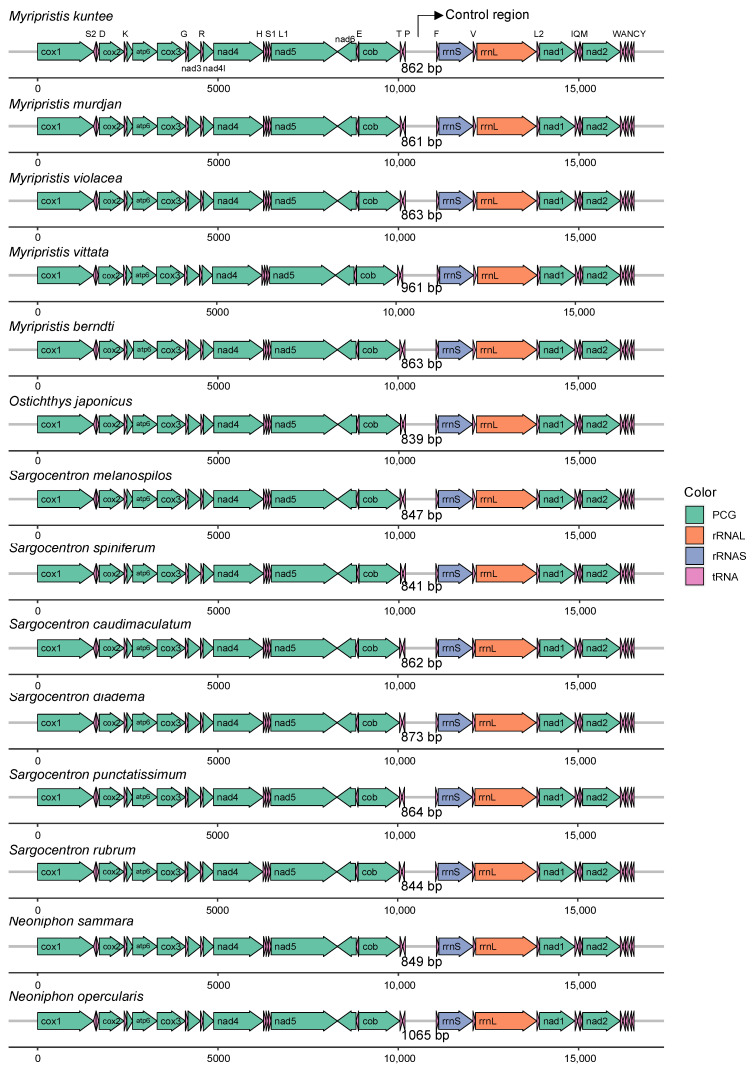
Gene arrangement and comparative genome analysis of 14 mitochondrial genomes of holocentrid species.

**Figure 3 biology-12-01273-f003:**
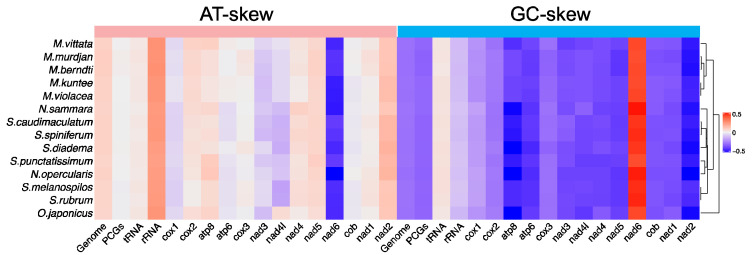
The base skew of various datasets among 14 mitogenomes, with hierarchical clustering of holocentrid species (*y*-axis) based on AT skew and GC skew.

**Figure 4 biology-12-01273-f004:**
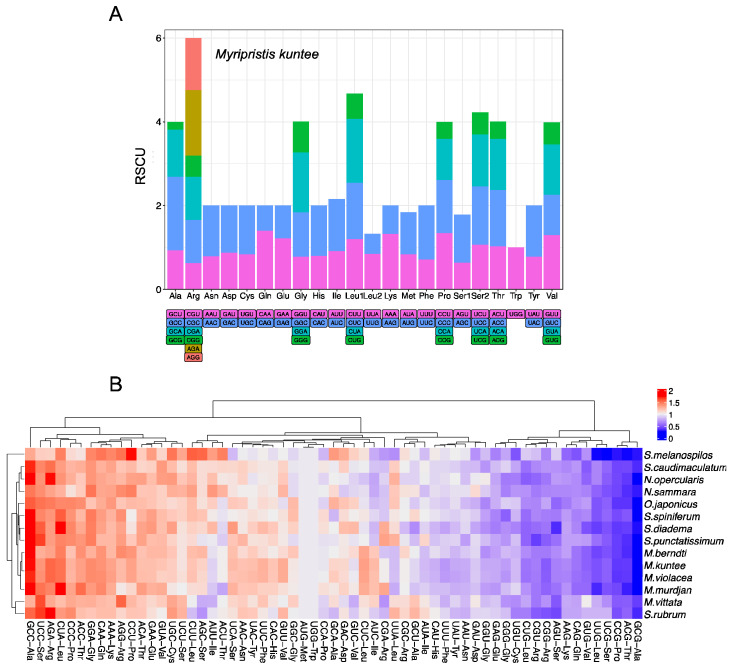
(**A**) Relative synonymous codon usage (RSCU) in the holocentrid mitogenome, for which *Myripristis kuntee* was taken as an example. Codon families are indicated below the *x*-axis. (**B**) Heatmap based on RSCU of 14 mitogenomes.

**Figure 5 biology-12-01273-f005:**
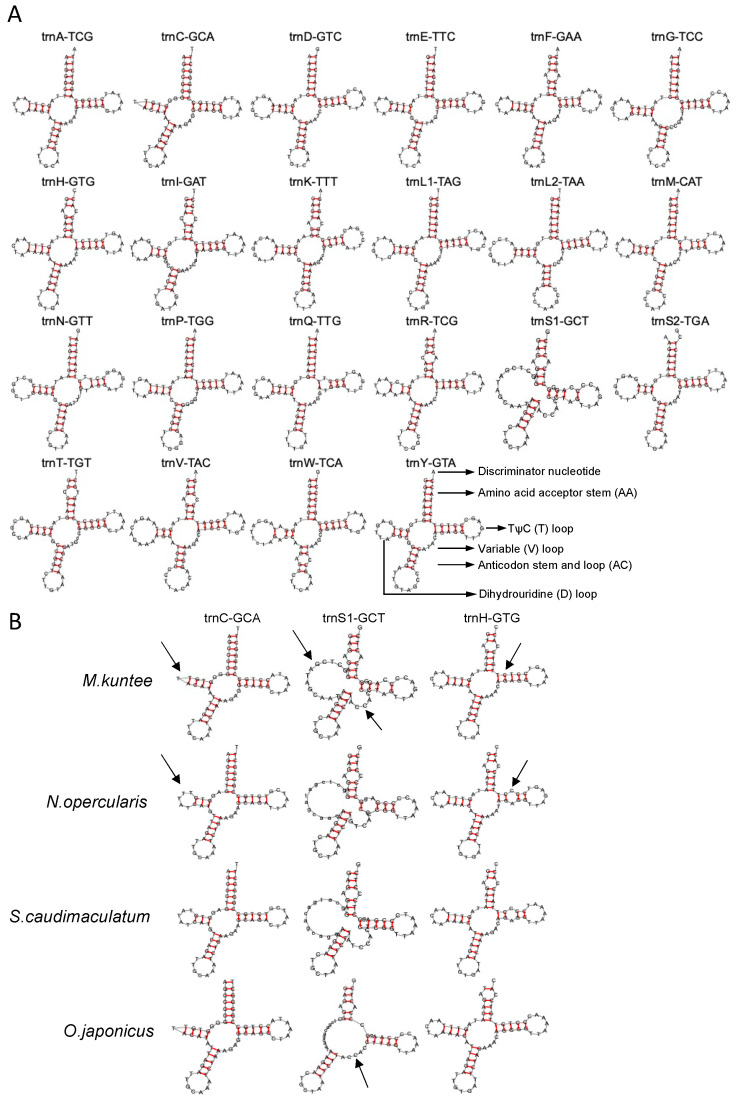
(**A**) Putative secondary structure of tRNAs in holocentrid mitogenomes. (**B**) Differential secondary structure of *trnC*-GCA (Ala), *trnS1*-GCT (Ala), and *trnH*-GTG (Val) of four holocentrid species. Arrows were used to highlight the different structural features of the three tRNAs.

**Figure 6 biology-12-01273-f006:**
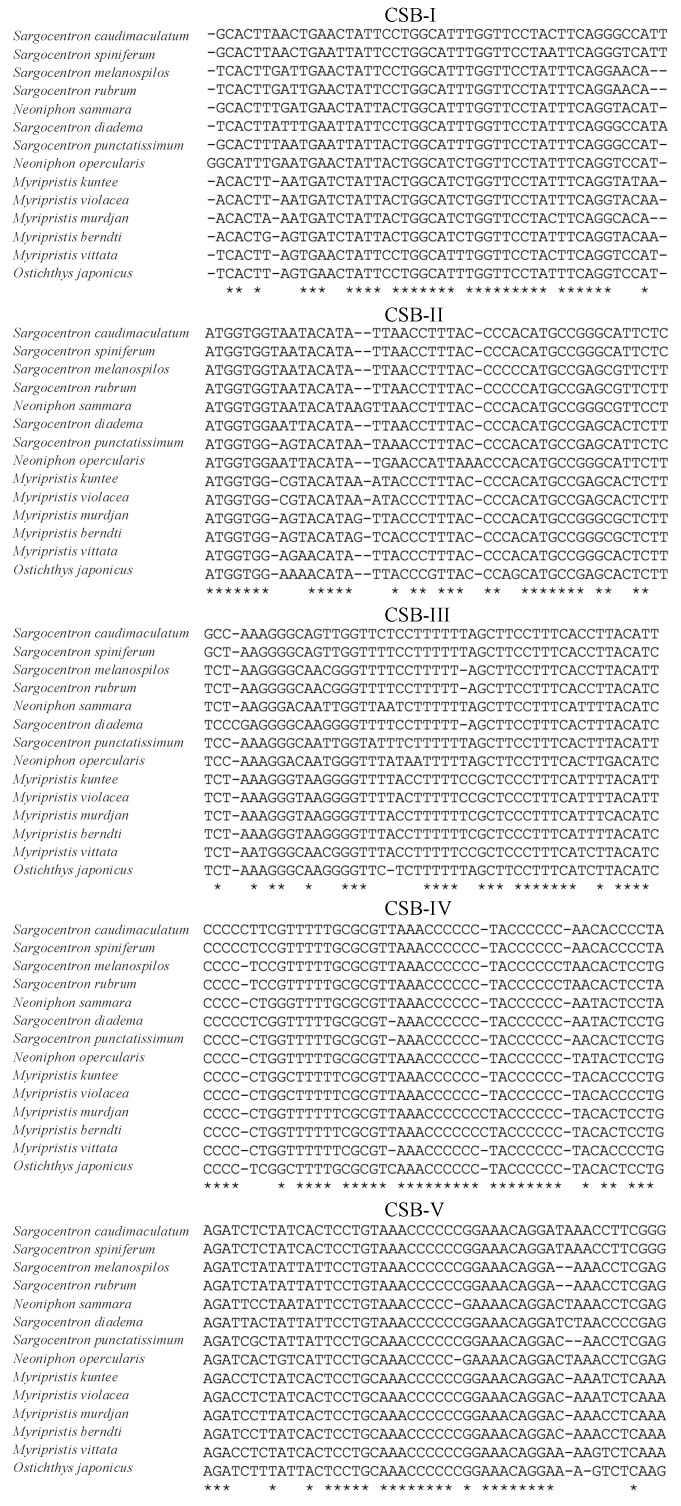
Conserved sequence blocks (CSBs) of the control region in the holocentrid mitogenomes. The asterisks are used to indicate the conserved sites.

**Figure 7 biology-12-01273-f007:**
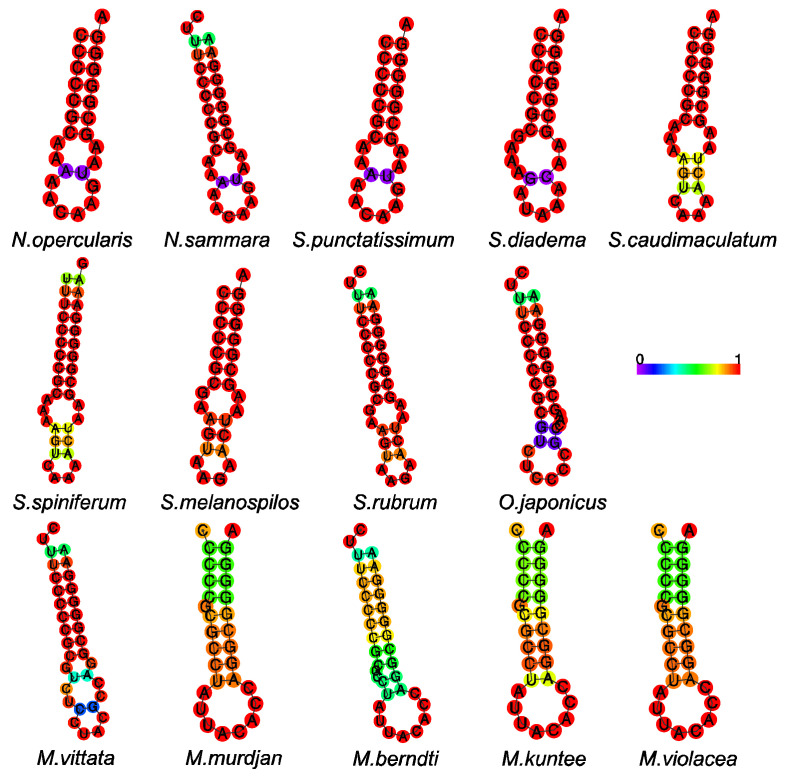
Putative secondary structure of the origin of L strand replication (O_L_) in 14 holocentrid species. The color-scaled bar represents base pairing probabilities calculated using CentroidFold software (http://rtools.cbrc.jp/centroidfold/, accessed on 1 September 2023).

**Figure 8 biology-12-01273-f008:**
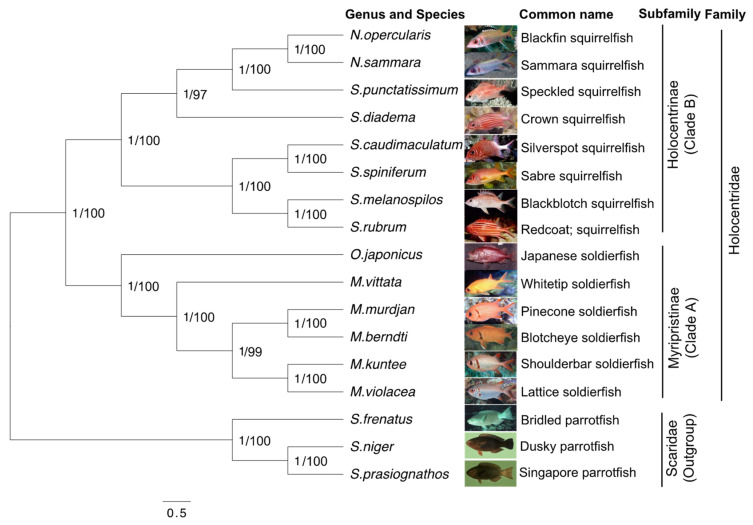
Phylogenetic trees were constructed using the nucleotide sequences of the 13 PCGs of the holocentrid mitogenomes, and three parrotfish mitogenomes were taken as the outgroup. Both the BI method and the ML method were used. The numbers beside the nodes are posterior probabilities (BI) and bootstraps (ML), respectively.

**Figure 9 biology-12-01273-f009:**
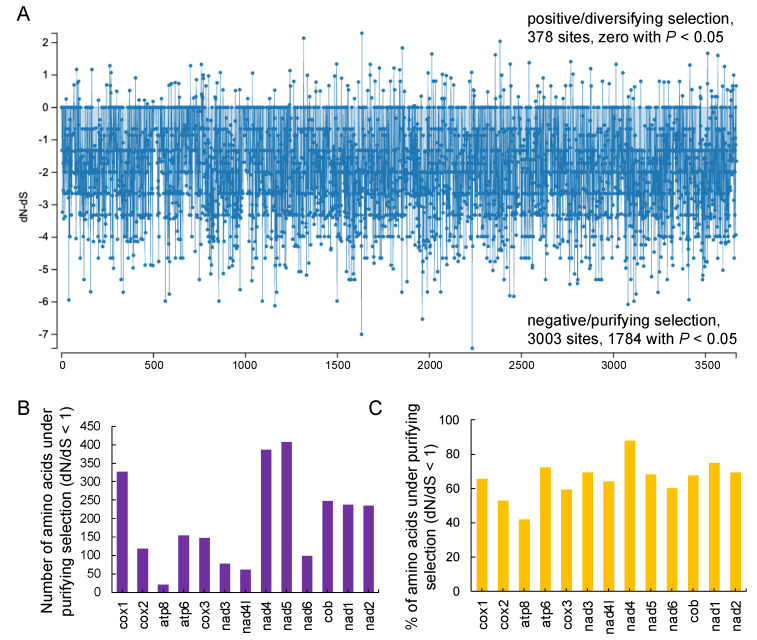
Positive selection of the 13 PCGs in the mitogenomes. (**A**) Detailed site-by-site dN − dS results from the SLAC analysis. (**B**) Number of amino acids under purifying selection (dN/dS < 1 or ω < 1) determined via the FEL method. (**C**) % of amino acids under purifying selection determined using the FEL method.

**Table 1 biology-12-01273-t001:** Summary of the mitochondrial genomes of holocentrid fish. *Myripristis kuntee* was taken as an example.

Gene	Start	End	Strand	Size (bp)	Start Codon	Stop Codon	Anticodons
*cox1*	1	1557	+	1557	GTG	AGA	−
*trnS2*	1553	1623	−	71	−	−	TGA
*trnD*	1627	1698	+	72	−	−	GTC
*cox2*	1712	2402	+	691	ATG	T	−
*trnK*	2403	2475	+	73	−	−	TTT
*atp8*	2477	2644	+	168	ATG	TAA	−
*atp6*	2635	3318	+	684	ATG	TAA	−
*cox3*	3318	4103	+	786	ATG	TAA	−
*trnG*	4103	4173	+	71	−	−	TCC
*nad3*	4174	4524	+	351	ATG	TAG	−
*trnR*	4523	4591	+	69	−	−	TCG
*nad4l*	4592	4888	+	297	ATG	TAA	−
*nad4*	4882	6262	+	1381	ATG	T	−
*trnH*	6263	6331	+	69	−	−	GTG
*trnS1*	6332	6399	+	68	−	−	GCT
*trnL1*	6401	6473	+	73	−	−	TAG
*nad5*	6474	8312	+	1839	ATG	TAA	−
*nad6*	8308	8829	−	522	ATG	AGG	−
*trnE*	8830	8898	−	69	−	−	TTC
*cob*	8905	10,045	+	1141	ATG	T	−
*trnT*	10,046	10,117	+	72	−	−	TGT
*trnP*	10,122	10,191	−	70	−	−	TGG
*trnF*	11,054	11,121	+	68	−	−	GAA
*rrnS*	11,122	12,070	+	949	−	−	−
*trnV*	12,071	12,142	+	72	−	−	TAC
*rrnL*	12,170	13,814	+	1645	−	−	−
*trnL2*	13,839	13,912	+	74	−	−	TAA
*nad1*	13,913	14,887	+	975	ATG	TAA	−
*trnI*	14,892	14,961	+	70	−	−	GAT
*trnQ*	14,961	15,031	−	71	−	−	TTG
*trnM*	15,031	15,100	+	70	−	−	CAT
*nad2*	15,101	16,147	+	1047	ATG	TAA	−
*trnW*	16,147	16,219	+	73	−	−	TCA
*trnA*	16,221	16,289	−	69	−	−	TGC
*trnN*	16,291	16,363	−	73	−	−	GTT
*trnC*	16,397	16,461	−	65	−	−	GCA
*trnY*	16,462	16,529	−	68	−	−	GTA

**Table 2 biology-12-01273-t002:** The base composition of the CSBs of the control region of parrotfish mitogenomes.

Base Composition (%)	CSB-I	CSB-II	CSB-III	CSB-IV	CSB-V
A	27.66	25.53	18.37	12.77	36.73
T	38.30	25.53	42.86	23.40	14.29
G	14.89	17.02	16.33	10.64	12.24
C	19.15	31.91	22.45	53.19	36.73

## Data Availability

The newly generated mitogenome sequences from the current study were deposited in the NCBI nucleotide database under the accession numbers OR148894-OR148901.

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
