# Peer review of "Comparative Mitogenome Analyses Uncover Mitogenome Features and Phylogenetic Implications of the Reef Fish Family Holocentridae (Holocentriformes)"

_biology, 2023, doi:10.3390/biology12101273_

Round 1

Reviewer 1 Report (Previous Reviewer 2)

Thank you for making the appropriate changes to the phylogenetic analysis.

Reviewer 2 Report (Previous Reviewer 3)

Authors have incorporated all the suggestions, accordingly. I recommend this article to publish in current version.

Reviewer 3 Report (Previous Reviewer 4)

Considering the efforts made by the Authors in revising and submitting this document following my previous comments and suggestions, as well as adding a more in-depth analysis of the control region, I have no other comments in this regard.

Best regards

The Reviewer

This manuscript is a resubmission of an earlier submission. The following is a list of the peer review reports and author responses from that submission.

Round 1

Reviewer 1 Report

After reviewing the manuscript titled "Comparative Mitogenome Analyses Uncover Mitogenome Features and Phylogenetic Implications of the Reef fish Family Holocentridae (Holocentriformes) " the authors investigated the mitogenome of eight species within the family Holocentridae. They conducted comparative mitogenome analyses with six other holocentrid species. I found there are some mistakes as well as concept misunderstandings in this manuscript, despite the authors spending tremendous efforts to decipher the phylogenetic relationship of Holocentridae with comprehensive tools. The described aims are not clear, and any hypotheses were proposed. The introduction section is dreadful and looks as though it has been written by an inexperienced team member, without scrutiny by the rest of the team. The introduction section will need to be rewritten to achieve clarity and give the reader the confidence that the manuscript is going to be worth reading. The author should have revised and described the phylogeny of family Holocentridae. It is really difficult for someone not familiar with the phylogeny of family Holocentridae to understand this paper and get the main idea of the paper. The described aims were not clear, and any hypotheses were proposed. In the Methodology section, there are some mistakes in the phylogenetic analysis. The methodology of genome sequencing must be detailed. The author used the HYPHY software for performing positive selection analysis. However, the author did not explain why this analysis was conducted or its implications. Instead, it seems more like a list of observations describing certain phenomena. In results and discussion section, the author must be improved by removing general statements and explaining really the results obtained and comparing these results with the other species in family Holocentridae. Their results need to be discussed from a phylogenetic perspective. Despite the comprehensive data presented the results and discussion were not deeply discussed. While reading the manuscript I encountered several issues that need the most careful attention from the authors. Some issues are semantic/conceptual others are technical. I am pleased to inform you that the following paper will not be officially accepted for publication until editing of the English grammar and phrasing by native English speaker. I tried making my review as constructive as possible and hoped the authors find it useful. Unfortunately, while I consider most sections of the article very poorly organized in their current form, in my opinion, this manuscript does not meet the criteria for publication and must therefore be a major revision.

Minor comments

1.     L56-L69, In Introduction section, we suggest the author to emphasize more on the description of the phylogeny of Family Holocentridae to help readers understand the phylogenetic implications.

2.     L70-84, The author must clarify and enumerate these previous studies that are unrelated to the mitochondrial whole-genome structure to help readers understand the significance of the research objectives.

3.     L85-L91, This is common sense, the author must revise it.

4.     L110-115, I suggest that the authors re-write these sentences based on the 'hypothesis and test' structure—clear questions raised in the introduction, then based on the question, to collect data and to do relevant analyses rather than every analysis, then in the discussion part, go back to the question defined in the introduction. The author's title is to explore the phylogenetic implications of the reef fish Family Holocentridae (Holocentriformes). The author should mention the phylogeny of Holocentridae in detail.

5.     L191-L202, In this section, the author describes and performs a positive selection analysis. We expected this analysis to be well-explained; however, from the author's results and discussions, it appears to be rather superficial. The author fails to provide a clear explanation of the significance of this analysis. We urge the author to make significant revisions to this section.

6.     L351-L353, The species studied by the author are fish. However, it is unclear why the example of sea cucumber was chosen for explanation.

7.     L372-L387, The phylogenetic analysis should be the most crucial part of this study, but the author seems to have overlooked this aspect. There is insufficient discussion on the phylogenetic relationships, particularly on how using the mitochondrial whole-genome as a genetic marker addresses certain questions related to phylogenetic evolution. We recommend the author to provide more emphasis on this section.

8.     L203-L210, The resolution of Figure 3 and Figure 4 is too low, and the author must improve the resolution of these two figures to help readers understand these signaling pathways.

9.     L396-L397, “purifying selection on PCGs makes sure the deleterious mutations cannot take over the population.”. There is the need to add references here to support the argument.

The author should add the reference.

10.  L399-L419, In Conclusions section, we suggest that the authors re-write these sentences based on the 'hypothesis and test' structure—clear questions raised in the introduction, then based on the question, to collect data and to do relevant analyses rather than every analysis, then in the discussion part, go back to the question defined in the introduction. The author's discussion in the abstract and conclusion is too similar. We recommend the author to revise the conclusion to enable readers to clearly understand the important findings and results of this research.

Figure 2, The author listed a diagram in Figure 2 showing the arrangement of 14 mitochondrial full-length gene sequences. However, I am having difficulty discerning the differences between these 14 gene sequences. I kindly request the author to provide further annotations or labels in the figure to aid readers in understanding these distinctions.

Figure 6, The phylogenetic tree in Figure 6 appears to have an error in the selection of outgroup species. We recommend the author to avoid choosing species from the same genus as outgroup representatives. Instead, it is more appropriate to select species with more distant phylogenetic relationships as outgroups.

Moderate editing of English language required

Author response

Major revision:
Reply: Thank you for your comments. We agree with the errors you pointed out and have
made major revisions according to your comments. Here is a point-by-point response to
your questions:
1. The described aims are not clear, and any hypotheses were proposed.
Reply: We revised the Simple Summary and Abstract to convey our design and
organization clearly. Firstly, we found similarities in morphology and distribution of two
subfamilies of Holocentridae but varying inhabitation and feeding behavior (Line 17 - Line
19). We assumed that phylogenetic relationships are consistent with the characteristics
of inhabitation and behavior. Then verified our hypothesis through comparative analysis
(Line 23 - Line 24). And in the Abstract section, we raised that the phylogeny, molecular
mechanisms, and adaptive strategies are unclear (Line 29). Methodologically, we
sequenced the mitogenomes and did the comparative analysis to reveal the molecular
differences among species (Line 29 - Line 36). Finally, we revealed the phylogeny and
selective strategies of species (Line 36 - Line 41).

All revised sentences are highlighted in red color in the revised manuscript.

2. There are some mistakes as well as concept misunderstandings on the phylogenetic
relationship of Holocentridae. The author should have revised and described the
phylogeny of family Holocentridae. It is really difficult for someone not familiar with the
phylogeny of family Holocentridae to understand this paper and get the main idea of the
paper. Their results need to be discussed from a phylogenetic perspective.
Reply: In the previous manuscript, we had a misunderstanding regarding the concept of
the “outgroup” in the phylogenetic tree. We incorrectly used species from the same family
as the outgroup, which is entirely incorrect. In the revised manuscript, we reconstructed
the phylogenetic tree by adding species from the other family as an outgroup. Specifically,
we included parrotfish from the coralfish family Scaridae, which had been previously
published by our corresponding author Teng Wang. Accordingly, we have revised the
methods and results sections (Line 173 - Line 175, and Line 357 - Line 371, Figure 8).
Besides, we revised the discussion and the conclusion sections and discussed more
about the phylogenetic relationship (Line 425 - Line 427, Line 439 - Line 447, Line 470 -
Line 474).
3. The introduction section is dreadful and looks as though it has been written by an
inexperienced team member, without scrutiny by the rest of the team.
Reply: The introduction section has been read sentence by sentence by the rest of the
team and revised by experienced team members. Please find the revisions at Line 51 -
Line 54, Line 67 - Line 69, Line 70 - Line 74, Line 82 - Line 84, Line 85 - Line 106, Line
107 - Line 114.
4. The author used the HYPHY software for performing positive selection analysis.
However, the author did not explain why this analysis was conducted or its implications.
Reply: We explained the reason and implications in the revised manuscript. At Line 195
- Line 198, Line 203 - Line 209, and Line 448 - Line 451.
5. In results and discussion section, the author must be improved by removing general
statements and explaining really the results obtained and comparing these results with
the other species in family Holocentridae.
Reply: Although the mitogenome structures are relatively conserved within the
Holocentridae family from our previous analysis, we further compared the 14 species in
the revised manuscript (Please find the results in the newly added section 3.4). We
detected the overlaps between genes (Line 321 - Line 327), analyzed the control regions
(Line 327 - Line 335, Figure 6 and Supplemental Figure 2), and compared the secondary
structure of the origin of L strand replication (OL) (Line 336 - Line 344, Figure 7). We
found variations in control regions and also identified five conserved sequence blocks
(CSB) (Table 2, Figure 6). Besides, we discussed the species more on phylogenic
analysis (Line 441 - Line 447).
Finally, we removed general statements, went over the entire manuscript and made
corrections to writing (or grammar) errors.
Minor revisions:
1. L56-L69, In Introduction section, we suggest the author to emphasize more on the
description of the phylogeny of Family Holocentridae to help readers understand the
phylogenetic implications.
Reply: Thank you for your suggestion. Previous literature only described morphological
differences and classifications of holocentroid fish. This study is the first time to
emphasize the evolutionary relationships of Holocentridae from the mitochondrial level.
So far, we haven’t found literature that studies phylogeny at the genomic and
mitochondrial levels. However, we found research that explored the evolution of
holocentroids from fossil materials. The general conclusion from the research has been
described in Line 76 - Line 80 of the revised manuscript.
2. L70-84, The author must clarify and enumerate these previous studies that are
unrelated to the mitochondrial whole-genome structure to help readers understand the
significance of the research objectives.
Reply: Thank you for your proposal. We emphasized the previous studies are unrelated
to the mitochondrial whole-genome structure and the following studies on mitogenomes
will reveal the systematic relationship of holocentrid species. Line 67 - Line 69.
3. L85-L91, This is common sense, the author must revise it.
Reply: Thank you for your suggestion. We think this common sense is necessary for new
people to understand mitochondria structure, especially people who have never studied
mitochondria before.
4. L110-115, I suggest that the authors re-write these sentences based on the
'hypothesis and test' structure—clear questions raised in the introduction, then based on
the question, to collect data and to do relevant analyses rather than every analysis, then
in the discussion part, go back to the question defined in the introduction. The author's
title is to explore the phylogenetic implications of the reef fish Family Holocentridae
(Holocentriformes). The author should mention the phylogeny of Holocentridae in detail.
Reply: Thank you for your good suggestion. We have improved the introduction part and
revised the Simple Summary and Abstract for the phylogeny part to highlight our findings
in red color. Finally, our tree supports the classification of holocentroids described in the
introduction (Figure 8), which are based on morphological research.
5. L191-L202, In this section, the author describes and performs a positive selection
analysis. We expected this analysis to be well-explained; however, from the author's
results and discussions, it appears to be rather superficial. The author fails to provide a
clear explanation of the significance of this analysis. We urge the author to make
significant revisions to this section.
Reply: Thank you for your comment. We have made significant revisions to this section.
Please see Line 195 - Line 216 in the revised manuscript.
6. L351-L353, The species studied by the author are fish. However, it is unclear why
the example of sea cucumber was chosen for explanation.
Reply: Thank you for your question. The purpose of referencing sea cucumbers is to
highlight the distinctive codon usage in bony fish.
7. L372-L387, The phylogenetic analysis should be the most crucial part of this study,
but the author seems to have overlooked this aspect. There is insufficient discussion on
the phylogenetic relationships, particularly on how using the mitochondrial whole-genome
as a genetic marker addresses certain questions related to phylogenetic evolution. We
recommend the author to provide more emphasis on this section.
Reply: Thank you for your question. We have further discussed it in the revised
manuscript. Line 440 - Line 447.
8. L203-L210, The resolution of Figure 3 and Figure 4 is too low, and the author must
improve the resolution of these two figures to help readers understand these signaling
pathways.
Reply: Maybe this is a random question. Because there is no Figure under L203-L210,
no signaling pathways in any figure of our manuscript, we didn’t mention any signaling
pathways in the whole manuscript. All our images achieved the pixel requirements.
9. L396-L397, “purifying selection on PCGs makes sure the deleterious mutations
cannot take over the population.”. There is the need to add references here to support
the argument. The author should add the reference.
Reply: Thank you for your question. We have added references to support the argument.
See Line 451.
10. L399-L419, In Conclusions section, we suggest that the authors re-write these
sentences based on the 'hypothesis and test' structure—clear questions raised in the
introduction, then based on the question, to collect data and to do relevant analyses rather
than every analysis, then in the discussion part, go back to the question defined in the
introduction. The author's discussion in the abstract and conclusion is too similar. We
recommend the author to revise the conclusion to enable readers to clearly understand
the important findings and results of this research.
Reply: Thank you for your comment. We have revised the conclusion to enable readers
to clearly understand the important findings and results of our research. Line 461 - Line
478, which is highlighted in red color.
Figure 2, The author listed a diagram in Figure 2 showing the arrangement of 14
mitochondrial full-length gene sequences. However, I am having difficulty discerning the
differences between these 14 gene sequences. I kindly request the author to provide
further annotations or labels in the figure to aid readers in understanding these
distinctions.
Reply: Thank you for your question. We have added a transparent box in Figure 2 to
indicate the difference between 14 mitogenomes, along with a supplementary figure to
support it (Supplementary Figure 2).
Figure 6, The phylogenetic tree in Figure 6 appears to have an error in the selection of
outgroup species. We recommend the author to avoid choosing species from the same
genus as outgroup representatives. Instead, it is more appropriate to select species with
more distant phylogenetic relationships as outgroups.
Reply: Thank you for your question. We revised the phylogenetic tree using three
5
parrotfish from the Scaridae family as an outgroup. And revised the methods part for the
construction of the phylogenetic tree (See Figure 8 in the revised manuscript).

Reviewer 2 Report

Your work with the analyzing the mitogenomes is handled well, but your phylogenetic analysis is entirely inappropriate. First, you cannot use members of your ingroup (in this instance Holocentridae) as your outgroup. You must re-root the tree with at least one, but preferably several other non-holocentrid fish species. I expect there are several appropriate choices out there with full mitogenomes to use. Second, your taxon coverage is inappropriate for the conclusions that you're making. You leave out one genus of Holocentrinae, and three genera from Myripristinae. Without including those, you can't really say anything substantive about the monophyly of either subfamily. Finally, the phylogenetic information that you present is no different than what you cover in the first paragraph of the introduction as past research, so you are not adding anything new here.

I suggest either removing the phylogenetic work entirely, or re-running it with actual outgroups and better taxonomic representation. Otherwise, the rest of the paper is very well written and presented.

Author Response

Reply: Thank you for your question. Firstly, we have re-run the phylogenetic tree using three parrotfish from the Scaridae family as an outgroup. The clades were reorganized and the classification of subfamilies became more rational. And accordingly revised the methods part for the phylogenetic analysis. Please find the revision at Line 173 - Line 175, Line 357 - Line 371, Line 439 - Line 447, and the final tree at Figure 8. Our new findings from the phylogenetic tree include: 1) the relationship between genera within each subfamily; 2) the phylogenetic position between subfamilies, not only the classification of genera described in the first paragraph of the introduction.

Reviewer 3 Report

Manuscript "Comparative Mitogenome Analyses Uncover Mitogenome Features and Phylogenetic Implications of the Reef fish Family Holocentridae (Holocentriformes)" is very interesting.

Authors reported eight mitogenomes for the first time and conducted a comparative analysis with six published mitogenomes. Authors revealed the detailed features of all mitogenomes on structure, gene arrangement, nucleotide composition, noncoding RNA, and codon usage. Authors investigated the phylogenetic relationships among these species and estimated selection pressure during the evolution.

Figure 3 needs a comparative analysis between AT-skew and GC-skew.

Figure 4: What method was used to calculate similarity? What method was used to construct the dendrogram?

Figure 6: What method was used to calculate similarity? What method was used to construct the dendrogram?

Paper needs minor revision.

Author Response

Authors reported eight mitogenomes for the first time and conducted a comparative analysis with six published mitogenomes. Authors revealed the detailed features of all mitogenomes on structure, gene arrangement, nucleotide composition, noncoding RNA, and codon usage. Authors investigated the phylogenetic relationships among these
species and estimated selection pressure during the evolution.

Figure 3 needs a comparative analysis between AT-skew and GC-skew.
Reply: Thank you for your question. We added a comparative analysis between ATskew and GC-skew, please see the first paragraph of section 3.2 in the revised manuscript (Line 263 - Line 265, and Line 272 - Line 274).

Figure 4: What method was used to calculate similarity? What method was used to construct the dendrogram?
Reply: Thank you for your question. We used ClustalW to calculate similarities. And used the K-means method from the R package “ComplexHeatmap” to draw the dendrogram. We have added descriptions of these methods in the methodology part (Line 168 - Line 169).

Figure 6: What method was used to calculate similarity? What method was used to construct the dendrogram?
Reply: Thank you for your question. We used ClustalW to calculate similarities. IQTREE and MrBayes were used to construct the phylogenetic tree, and FigTree software was used to visualize the tree as a dendrogram. We have mentioned it in the method section (Line 192 - Line 193).

Reviewer 4 Report

I found this manuscript interesting based on a well-conducted study on an important family of reef corals fish. The analyses and results are interesting, especially the ones related to the PCGs, as expected. The main limitation of this study is the absence of an investigation conducted on the control regions of the sequenced mtDNAs, which is increasing in importance in genomic research of aquatic organisms in the last period, from multiple points of view. Could the authors provide our ideas about this, or provide more information about it within the manuscript?

Considering the Journal guidelines and the contents of sections, I suggest inverting Simple Summary and Abstract. Indeed, the first one is more wordy and accurate than the Abstract, resulting in less detail.

Please avoid using the Keywords of words already reported in the Title.

The mtDNA maps of all the newly sequenced species should be added to the supplementary material as useful references.

Best regards

The Reviewer

Author Response

Considering the Journal guidelines and the contents of sections, I suggest inverting Simple Summary and Abstract. Indeed, the first one is more wordy and accurate than the Abstract, resulting in less detail.

Reply: Thank you for your comment. We have revised the Simple Summary and Abstract, and have inverted part of the results. Please find the changes in Simple Summary at Line 24 – Line 26, and at Line 36 – Line 40 in the Abstract, these two parts were reorganized and inverted.

Please avoid using the Keywords of words already reported in the Title. 

Reply: Thank you for your comment. We only used two words, “Holocentridae” and “mitogenome” in the Keywords part, which is also in the Title. These two words are real keywords of the whole paper, we think they should not be removed.

The mtDNA maps of all the newly sequenced species should be added to the supplementary material as useful references.
Reply: Thank you for your comment. We have added them to the supplementary materials,
please see supplementary Figure 1.